# Robust Scheduling of Multi-Skilled Workforce Allocation: Job Rotation Approach

Eryk Szwarc [1,*], Paulina Golińska-Dawson [2], Grzegorz Bocewicz [1] and Zbigniew Banaszak [1]

1   Faculty of Electronics and Computer Science, Koszalin University of Technology, Sniadeckich 2,
    75-453 Koszalin, Poland; grzegorz.bocewicz@tu.koszalin.pl (G.B.); zbigniew.banaszak@tu.koszalin.pl (Z.B.)
2   Faculty of Engineering Management, Poznan University of Technology, Jacka Rychlewskiego 2,
    60-965 Poznań, Poland; paulina.golinska@put.poznan.pl
*   Correspondence: eryk.szwarc@tu.koszalin.pl

**Abstract:** This paper addresses scheduling challenges in software development organizations, specifically focusing on a novel version of the software project scheduling problem (SPSP). This enhanced model incorporates the dynamics of learning and forgetting phenomena, crucial in maintaining employee competencies, particularly when unexpected events such as absenteeism or shifts in project priorities occur. The paper introduces a new declarative reference model for SPSP, aimed at proactively managing the assignment of versatile programmers to tasks within an portfolio of IT projects, while considering the effects of forgetting. Implemented within a constraints programming environment, this model facilitates decision making in project management for software companies. It serves to find feasible solutions and identify conditions necessary to meet specified expectations. The effectiveness of the proposed SPSP model is demonstrated through numerical examples.

**Keywords:** job rotation; project; competencies; maintenance; learning–forgetting effect; robustness

## 1. Introduction

In today's dynamic and competitive business environment, organizations are constantly seeking ways to optimize their resources, enhance productivity, and adapt to changing demands. A critical aspect of achieving these goals is efficient scheduling of a multi-skilled workforce allocation [1–4]. A promising solution is an effective approach that results in the job rotation driven robust scheduling of employees, which not only optimizes the allocation of a workforce but also promotes an employee's skill development and job satisfaction [5,6].

Instead of confining employees to a single job profile, job rotation (JR) involves the systematic movement of employees between different roles and functions within an organization, encouraging them to take on varied responsibilities, allowing for the development of a broad skillset [7–11]. Implementing a JR approach in workforce scheduling addresses these challenges by providing a dynamic framework that adapts to changing demands while allowing for real-time adjustments in workforce allocation based on demand fluctuations. This ensures optimal resource utilization without compromising efficiency while balancing the organizational needs with the capabilities of individual employees [12–15].

Businesses often face fluctuations in demand, requiring dynamic scheduling algorithms that consider real-time factors such as employee availability and skill requirements. Coordinating a workforce with diverse skillsets following such a factor requires careful planning caused by the effectiveness of job rotation programs being influenced by the learning–forgetting effect, which suggests that the skills acquired during a rotation can be diminished if not consistently reinforced [16–18]. This means that multi-skill employees in the professions of a teacher (conducting classes in various subjects), an IT programmer (using different programming languages), or a nurse (working in emergency, surgical,

and pediatric wards) deteriorate over time if not reinforced, i.e., when their periodic, alternating use is not entailed. In other words, the learning–forgetting effect introduces a temporal dimension to this maintenance challenge, emphasizing the need for continual skill reinforcement to counteract the natural decay of knowledge over time. Against this background, the presented study explores the learning–forgetting effect and its implications on the scheduling of job rotations to maintain the multi-skilled staff at an assumed level of their competences while involving the preservation and enhancement of a diverse skillset [19,20].

The purpose of this paper is to present a new decision-making model, which applies a declarative approach for the proactive planning of job rotation of a multi-skilled team of programmers to avoid the forgetting effect [21]. The work focuses on the skills of programmers, which are also called competencies. In software development, the level of the programmers' competencies (which changes over time due to job rotation) affects the execution time of tasks. Low-skilled programmers make more mistakes (requiring time-consuming corrections) compared to high-skilled programmers.

Therefore, this article represents an extension of previous research on proactive resource allocation within competency structure constraints [22–25] and is a continuation of the authors' recent research presented at the International Conference on Distributed Computing and Artificial Intelligence, 2023 [26].

The main contributions include the following:

- A new declarative model for proactive job rotation planning within a multi-skilled workforce.
- The definition of a constraint satisfaction software project scheduling problem (SPSP) that links staff rotation to task timing and competency validity. The new SPSP takes into account the uncertainties and dynamic events that often occur during the implementation of a software project.
- A demonstration through numerical results of the viability of employing this proactive job rotation planning method in real-time scenarios.

The structure of this paper is outlined as follows: Section 2 identifies the primary research gap based on the examination of related works. Section 3 introduces a declarative model that facilitates the formulation of the software project scheduling problem (SPSP) as a constraint satisfaction problem (CSP). Subsequently, in Section 4, the paper integrates the concepts of competence structure robustness and the maintenance of team competences at a constant level. The effectiveness of proactive job rotation schedules in SPSP is then revealed through computer experiments presented in Section 5. Finally, Section 6 summarizes the contributions of the paper.

## 2. Literature Review

The definitions of job rotation found in the literature and a review of research on job rotation in various types of activities, in particular in software producing enterprises, are presented. Following an exploration of software project scheduling within the programming industry, the research highlighted the influence of learning and forgetting effects. Through this investigation, a distinct gap in the existing body of knowledge was identified.

### 2.1. Job Rotation

Job rotation, also known as work rotation, represents a managerial tactic employed within organizations to alleviate work monotony, boredom, fatigue, and burnouts [27,28]. Its primary objective extends to improving job satisfaction, employee motivation and balancing ergonomics indices [29–31]. This strategy spans various sectors, encompassing nursing, engineering, software development, and the manufacturing industry [32,33]. In the context of software production, job rotation entails the deliberate and structured movement of employees across different job roles (J2J) or from one project to another (P2P) [34]. P2P rotation particularly involves shifting employees between projects that share similar characteristics, often involving comparable technical duties. This practice

stimulates greater team adaptability and reduces work monotony while also fostering innovation and the formation of diverse multicultural teams.

There are many definitions in the literature describing job rotation. In [27], it is treated as the deliberate and organized movement of employees within and between organizational areas to increase the company's success and the employment of employees. In [35], it was stated that it is a regular change of jobs in different positions in the organization, either based on a pattern or spontaneously based on the personal needs of employees. One study defines job rotation as the replacement of personnel between two or more areas of an organization for a predetermined period of time [30]. Another paper presents a practice that allows employees to be transferred from team to team and from project to project within the same organizational area [36]. One of the broadest definitions was proposed by Wood [37], who perceived job rotation as the systematic movement of employees from work to work or from project to project within an organization during the course of a task as an approach to achieving a wide variety of human resource goals, such as filling jobs, directing new employees, preventing boredom or burnouts, rewarding employees, and supporting career development.

In the context of software engineering organizations (software development and/or development), job rotation is defined as the practice of transferring an employee within an organization from one project (source project) to another (target project). In most situations, the role (software engineer, test engineer, team leader, software architect, etc.) performed by the rotated person remains the same. However, in different circumstances, in order to meet the resource needs in the target project, an employee can change the role, for example, a test engineer to a software engineer.

Typically, rotation is applied in the following two scenarios:

- When a particular project demands a larger team or a different skillset.
- When an employee expresses an interest in transitioning to different projects.

From this, it follows that rotations can be used to achieve organizational goals related to resource allocation, as well as individual motivational needs. One of the key factors to consider when performing a rotation is the variety of tasks and skills. Low diversity (shifting people to the same role in a project with similar technical requirements or to work in the same business domain using the same technologies) creates fewer opportunities for learning and therefore fewer benefits for the rotated person. On the other hand, too much variety can create a long and steep learning curve for a rotating person, which can result in a loss of performance and an increase in fatigue. Another important factor is the timing of the job rotation. Research indicates that a balance needs to be struck in a person's rotation frequency. Doing them too often reduces the perception of performing a job well. On the other hand, leaving a person in the same project for too long is not desirable because software engineers value variety (tasks and/or acquiring new skills). Overall, the research indicates that the use of rotation has a positive effect on motivation, job satisfaction, and innovation [38] and impacts negatively on job specificity [39]. A set of tasks and a set of employees with specific characteristics are included. Skills, remuneration, and working time limits are given. In the context of software projects, it is possible to have skills in specific technologies or tasks such as programming, database management, design, etc.

## 2.2. Scheduling of IT Projects

The rotation of programmers is part of the so-called software project scheduling problem (SPSP), which involves the assignment of employees to tasks, subject to various constraints, such as the total development cost and duration [40]. Many of the SPSP models proposed so far are similar to the model originally presented in [41], i.e., specified by a given set of tasks and a set of employees characterized by their skills, salary, preferences, working time limits, and so on. Skills are understood as the abilities possessed by employees to perform certain types of tasks. In the context of software projects, it is possible to have skills in specific technologies or tasks such as programming, database management, design, etc. In [40], an SPSP model is proposed that takes into account the uncertainties and dynamic

events that often occur during the implementation of a software project. In other words, the changing environment for software companies means that software project planning is a dynamic optimization problem. In this model, the labor intensity of the tasks is uncertain, that is, modifications to task specifications and inaccuracies in the initial estimates can cause changes in the originally estimated labor intensity of the task. Additionally, changes in customer requirements may occur during the software development lifecycle, leading to the need for additional tasks. Furthermore, the proposed model takes into account the fluctuation of employees (departure or employment) during the course of the project.

The existing SPSP models assume that the requirements for skills in the performance of tasks and the workload of employees are expressed in person–months. A task can be assigned to one or more employee, and each employee is assigned a level of commitment to the task. For example, if a full-time employee is assigned a commitment of 0.5 for a particular task, it means that that particular employee devotes half of his/her workday to that task. On this basis, the duration of the project is calculated.

The skills of programmers in SPSPs have been discussed in many studies so far. In [42], workers are classified according to four different skill levels, beginner, junior, senior, and expert, with an entry-level employee receiving the lowest salary and an expert earning the highest salary. Additionally, tasks are assigned a required skill level, which corresponds to one of the four mentioned. In [43], five different skill levels of workers are considered, i.e., novice, average, good, very good, and expert, while in [44], they are classified only as novice or experienced. In [45], employee productivity, i.e., the time it takes for an employee to complete their assigned tasks, was discussed.

In some SPSP models, the skill level of programmers is assumed to be constant [46,47]. In fact, employees can hone their skills during software development, i.e., an employee may have a low level of a skill, but if they work long enough on a task that requires that skill, they will increase the level of that skill through the experience they have gained. On the other hand, when they rarely use a particular skill, they experience a forgetting effect, and the skill level decreases [48], which affects the efficiency and working time of programmers and thus the timely completion of projects [49].

### 2.3. The Effect of Learning and Forgetting

The learning–forgetting effect is a cognitive phenomenon that occurs when newly acquired knowledge or skills gradually decline over time in the absence of reinforcement or practice. Understanding the dynamics of this effect is crucial to designing effective job rotation programs. Research suggests that the rate of forgetting varies depending on the complexity of the skills and the frequency of reinforcement activities.

Wright [50] was the pioneer in analyzing how the workers' skill levels impact the output of aircraft production, introducing the concept of the Wright learning curve (WLC). This curve illustrates the correlation between skill mastery or knowledge and time. It has been applied in the solution of project planning issues [51]. Additionally, studies such as [52] have demonstrated that utilizing the learning curve significantly influences the efficiency of the programmers' work.

The ability to learn is often accompanied by what is known as the forgetting effect—a non-linear function that represents the decline of a skill when it is not actively utilized. Several models have emerged to describe this phenomenon, such as the variable regression to variable forgetting (VRVF) model [53], the variable regression to invariant forgetting (VRIF) model [54], the learning–forgetting curve model (LFCM) [55], and the recency model (RC) [56].

While numerous studies exist in this domain, there is limited literature specifically addressing the learning and forgetting effects within SPSP. Existing research primarily delves into the impact on project costs and duration [18]. Notably, the focus tends to center on the learning effect, where programmers repeating the same tasks in a project's initial phase (without rotation) enhance specific skills, thereby reducing the implementation

time of those tasks in subsequent stages. However, this comes at the cost of diminishing proficiency in unused skills.

The study results do not definitively indicate whether repetitive task assignments without rotation and specialization in specific skills yield better outcomes than implementing work rotation. Additionally, within the realm of SPSP, there is a research gap concerning the influence of rotation and its associated learning and forgetting effects on sustaining the employees' skill at consistent readiness levels. This aspect is important, especially when unforeseen disruptions occur during the functioning of software companies. To illustrate the problem of determining the schedules' robustness to disruptions, an example of an SPSP was used (see Section 3), in which, for didactic reasons, fictitious small-scale data were adopted.

## 3. Problem Formulation

The essence of the SPSP is the allocation of employees (hereinafter also referred to as programmers) with many skills to perform a specific set of tasks $Z = \{Z_1, \ldots, Z_i, \ldots, Z_Q\}$, such as the physical design and implementation of a database, user interface design, program coding, testing, etc. A portfolio of projects $E = \{E_1, \ldots, E_j, \ldots, E_K\}$ is given, where $E_j \subseteq Z$. The projects are executed according to the sequence $W = (w_1, .., w_l, \ldots, w_L)$, where $w_l \in E$.

A group of $\mathcal{P} = \{P_1, \ldots, P_k, \ldots, P_M\}$ programmers needed to carry out individual $Z_i$ tasks is given. The competence structure of employees $P$ during the execution of the $E_j$ project has the form of a matrix $G^j$ as follows:

$$G^j = \left[g_{i,k}^j\right]_{i=1\ldots Q; \, k=1\ldots M} \tag{1}$$

where $g_{i,k} \in \{1, \ldots, 5\}$ determines the level of competence of the programmer $P_k$ necessary to perform task $Z_i$. The adopted five-level scale determines the duration time $t_i^j$ of task $Z_i$ in accordance with the following function:

$$g_{i,k}^j = \begin{cases} 5 \text{ or } 4 & \text{programmer } P_k \text{ performs the task } Z_i \text{ in } t_i^j = 1 \text{ time unit} \\ 3 & \text{programmer } P_k \text{ performs the task } Z_i \text{ in } t_i^j = 2 \text{ time units} \\ 2 \text{ or } 1 & \text{programmer} P_k \text{performsthetask } Z_i \text{ in } t_i^j = 4 \text{ time units} \end{cases}$$

According to the above, the duration time of task $Z_i$ is:

- 1 time unit if the programmer $P_k$ has competences at level 4 or 5;
- 2 time units if the programmer $P_k$ has competences at level 3;
- 4 time units if the programmer $P_k$ has competences at level 2 or 1.

Each project $E_j$ from portfolio $E$ is associated with the $X^j$ assignment of programmers to tasks. Similar to the competence structure, it takes the form of a matrix, which is as follows:

$$X^j = \left[x_{i,k}^j\right]_{i=1\ldots Q; \, k=1\ldots M} \tag{2}$$

where $x_{i,k}^j \in \{0, 1\}$, $x_{i,k}^j = 1$ when the programmer $P_k$ is assigned to perform the task $Z_i$ in project $E_j$, otherwise, $x_{i,k}^j = 0$.

Assignments $X^j$ make up the sequence of assignments $X = \left(X^1, \ldots, X^j, \ldots, X^K\right)$, which determines the schedule for project execution as follows:

$$Y = (y_1, \ldots y_j, \ldots, y_K) \tag{3}$$

where $y_j$ specifies when project $E_j$ starts (all project tasks start with $E_j$ simultaneously). The completion time of project sequence order $W$ depends on schedule $Y$ and is denoted as variable $EY$.

According to the effect of learning and forgetting, the level of competence does not remain constant. Curves that model this effect are the subject of many studies [57,58], in which different shapes are given with different degrees of inclination. In this work, for simplicity, it was assumed that the level increases if the programmer performs a specific task. The range of this increase depends on the level of competence, and similarly, the level of competence decreases if the programmer does not perform a specific task. The classification rules are as follows:

- Competence increases by one for every two time units of task duration if the competency level is 1 or 2;
- Competence increases by one for each time unit of task duration if the competency level is 3 or 4;
- Competence decreases by one for each time unit when the task is not performed if the competence level is 2 or 3;
- Competence decreases by one for every two time units when the task is not performed if the competency level is 4 or 5.

In other words, the level of the programmers' competences changes during the implementation of subsequent projects. It depends on the assigned sequence adopted. The elements $g_{k,i}^j$ of the competence structure $G^j$ for project $E_j$ are determined as follows:

$$g_{i,k}^j = \varphi\left(g_{i,k}^{j-1}, x_{i,k}^{j-1}\right), \text{ for } j > 1, \tag{4}$$

where $\varphi$ is a function determining the level of competence of programmers for the $E_j$ project, based on the values of the elements $g_{i,k}^{j-1}, x_{i,k}^{j-1}$ (according to above rules).

In addition, the problem under consideration assumes the following:

- The task can be performed by only one programmer with any level of competence;
- All tasks must be assigned to employees;
- The moment $EY$ of completing sequence projects $W$ determined by the schedule $Y$ must be less than or equal to the arbitrarily set horizon $H$ ($EY \leq H$).

Unplanned events (disruptions) during the execution of projects mean adding a new project $^{(r)}E$ to the $W$ order (usually as the last one) as follows: $^{(r)}W = W||^{(r)}E$ (where $||$ means sequence concatenation operation). The ability of the competency structure to deal with this type of disruption can be assessed by the robustness measure (i.e., the ratio of variants of the number of disruption ($LP$) for which the competency structure $G$ guarantees the execution of $^{(r)}W$ to all disruption scenarios (set of disruptions $E^*$) as follows:

$$R = \frac{LP}{|E^*|}. \tag{5}$$

The problem outlined here simplifies the SPSP, overlooking various practical limitations, such as working hour constraints and payroll expenses. The presented approach focuses solely on the learning and forgetting curve model (LFCM), excluding other models such as RC (recency) and PID (power integration diffusion) for comparison—a direction for future investigation. In a broader context, diverse shapes of the forgetting function, specific to various professions, gender, and age groups, could be considered. However, due to the absence of available statistics defining these shapes for distinct occupational groups, this paper relies on approximate characteristics gathered from a population of 25–30 individuals over the last five years.

In this context, the objective is to find a response to the following specific questions:

1. Is there a particular assignment $X$ variation that guarantees the completion of the project's portfolio W within the specified horizon $EY \leq H$?
2. Does there exist an assignment $X$ that guarantees a specified robustness $R$ (e.g., $R = 1$) for completing tasks in the given project order?

To illustrate, let us consider a software company that executes an order involving a portfolio of three $E = (E_1, \ldots, E_3)$, projects that require different sets of steps (operations): $E_1:(Z_1, Z_2, Z_3)$; $E_2:(Z_1, Z_2, Z_4)$; and $E_3:(Z_2, Z_3, Z_4)$. Projects are completed in the following order: $E_1, E_2, E_3$, which means that only after the completion of project $E_1$, the tasks of project $E_2$, etc., begin.

The company employs three programmers $P = (P_1, \ldots, P_3)$. Each of them has a specific level of skills (on a scale of 1–5) to carry out individual tasks $Z = (Z_1, \ldots, Z_4)$. Their collective list is represented by the competence structure $G^1$ in Table 1 (structure describing the competencies of employees before $E_1$ project implementation). In addition, it is assumed that each employee has the 4th level of competence for each task.

**Table 1.** Competence structure $G^1$.

| $G^1$ | $P_1$ | $P_2$ | $P_3$ |
| --- | --- | --- | --- |
| $Z_1$ | 4 | 4 | 4 |
| $Z_2$ | 4 | 4 | 4 |
| $Z_3$ | 4 | 4 | 4 |

Programmer competency levels affect duration $t_i^j$ of tasks $Z_i$. To illustrate this, let us assume the following:

- Duration is $t_i^j = 1$ time unit if the level of competence of the employee is 4 or 5;
- Duration is $t_i^j = 2$ time unit if the level of competence of the employee is 3;
- Duration is $t_i^j = 4$ time unit if the level of competence of the employee is 2 or 1.

It was assumed that the level of competence increases if a programmer performs a specific task, and the rate of increase depends on the level of competence as follows:

- If the level of competency is 1 or 2, the competence increases by one per two time units of task duration;
- If the level of competency is 3 or 4, the competence increases by one for each time unit of task duration.

Similarly, the competence level decreases if the programmer does not perform a specific task, and the rate of decrease depends on the level of competence as follows:

- If the competence level is 2 or 3, the competence decreases by one for each time unit when the task is not performed;
- If the competency level is 4 or 5, the competence decreases by one in two time units of the time when the task is not performed.

For example, if a programmer $P_1$ with the competence to $Z_1$ at level 4 performs this task, then after 1 time unit, the level $Z_1$ will increase to 5. Similarly, if an employee does not perform a specific task, the level of competence decreases by one level. For example, if a programmer $P_2$ with the competence to $Z_1$ at level 4 does not perform this task in 2 time units, then the level will drop to 3. This means that at the implementation stage of each $E_j$ project, the levels of employees' competencies (described by the $G^j$ competency structure) may differ from their initial values.

For such assumptions, the sequence of assignments $X = (X^1, X^2, X^3)$ of the set of programmers $P$ to tasks $Z$ comprising projects $E_1, E_2, E_3$ is sought. The assignment $X^j$ determines the assignment of programmers (with competencies defined by the $G^j$ structure) to the tasks of the $E_j$ project.

It is required that each assignment $X^j$ meets the following conditions:

- The activity can be carried out only by one employee of any level of competence;
- All activities must be assigned to employees.

An example of assigning $X^1$ employees to tasks $Z_1, Z_2, Z_3$ in project $E_1$ is presented in the Table 2. A value of 1 means that the programmer is performing a specific task, and

a value of 0 means the opposite. For example, $P_1$ performs task $Z_1$ but does not perform tasks $Z_2$ and $Z_3$. It also does not carry out the task of $Z_4$, which is not part of the $E_1$ project.

**Table 2.** Employee assignment $X^1$.

| $X^1$ | $P_1$ | $P_2$ | $P_3$ |
|---|---|---|---|
| $Z_1$ | 1 | 0 | 0 |
| $Z_2$ | 0 | 1 | 0 |
| $Z_3$ | 0 | 0 | 1 |
| $Z_4$ | 0 | 0 | 0 |

The assignment $X^1$ defines both the $G^2$ competency structure for the subsequent implementation of the $E_2$ project as well as the time of completion of the commissioned tasks ($t_i^j$). For example, after the completion of the $E_1$ project, the level of $Z_1$ competence in the employee $P_1$ (learning effect) will increase (level 5), which means the time of implementation $t_1^2 = 1$, while the level of competence to $Z_2$, $Z_3$ and $Z_4$ will not change (level 4), which means the time of implementation $t_2^2, t_3^2, t_4^2 = 1$ (see Table 3).

**Table 3.** $G^2$ competence structure for the $E_2$ project (implemented after the end of the $E_1$ project).

| $G^2$ | $P_1$ | $P_2$ | $P_3$ |
|---|---|---|---|
| $Z_1$ | 5 ($t_1^2 = 1$) | 4 ($t_1^2 = 1$) | 4 ($t_1^2 = 1$) |
| $Z_2$ | 4 ($t_2^2 = 1$) | 5 ($t_2^2 = 1$) | 4 ($t_2^2 = 1$) |
| $Z_3$ | 4 ($t_3^2 = 1$) | 4 ($t_3^2 = 1$) | 5 ($t_3^2 = 1$) |
| $Z_4$ | 4 ($t_4^2 = 1$) | 4 ($t_4^2 = 1$) | 4 ($t_4^2 = 1$) |

The adopted allocation of employees $X$ determines the schedule for the implementation of projects $Y$. It is defined as the sequence $Y = (y_1, y_2, y_3)$, where $y_j$ specifies the start point of the $E_j$ project. For simplicity, it is assumed that all $E_j$ project tasks start at the moment of $y_j$. The start time of the $E_j$ project must not be less than the end time of the project $E_{j-1}$ (for $j > 1$).

The presented example illustrates a simplified version of the SPSP, which omits a number of limitations encountered in practice, such as working time limits, wage costs, etc. In general, however, the following routine question is sought: is there an assignment of $X$ employees that guarantees the existence of a schedule $Y$ for the implementation of projects $E_1, E_2, E_3$ that meets the given assumptions?

In general, there can be multiple assignment variants. Figure 1 shows the two selected variants of schedule $Y = (0, 1, 2)$ for two different assignments $X$. These schedules differ in terms of the time to complete the project portfolio. For example,

- In the variant from Figure 1a, assignments $X^1$ and $X^2$ cause a decrease in competences in the competence structures $G^2$ and $G^3$, affecting the duration of project $E_3$, i.e., in assignment $X^3$, employee $P_1$ (due to $G^3$, his $Z_4$ competence is at level 3) is assigned to task $Z_4$, which means that he performs this task in 2 time units ($t_4^3 = 2$). Ultimately, the project order ends after 4 time units.
- In the variant from Figure 1b, compared to the variant from Figure 1a, there is a higher rotation of tasks (another assignment $X^2$ causes a smaller decrease in competences in the $G^3$ structure after the implementation of project $E_2$), which allows project $E_3$ to be completed within 1 time unit. Ultimately, the project order ends after 3 time units.

Among the variants of schedule $Y$ that meet the adopted assumptions (see Figure 1), an answer to the following question is sought: which variant of employee assignment $X$ guarantees the minimum makespan to complete projects $E_1, E_2, E_3$?

In the example shown, such a variant of allocation $X$ and schedule $Y = (0, 1, 2)$ is illustrated in Figure 1b.

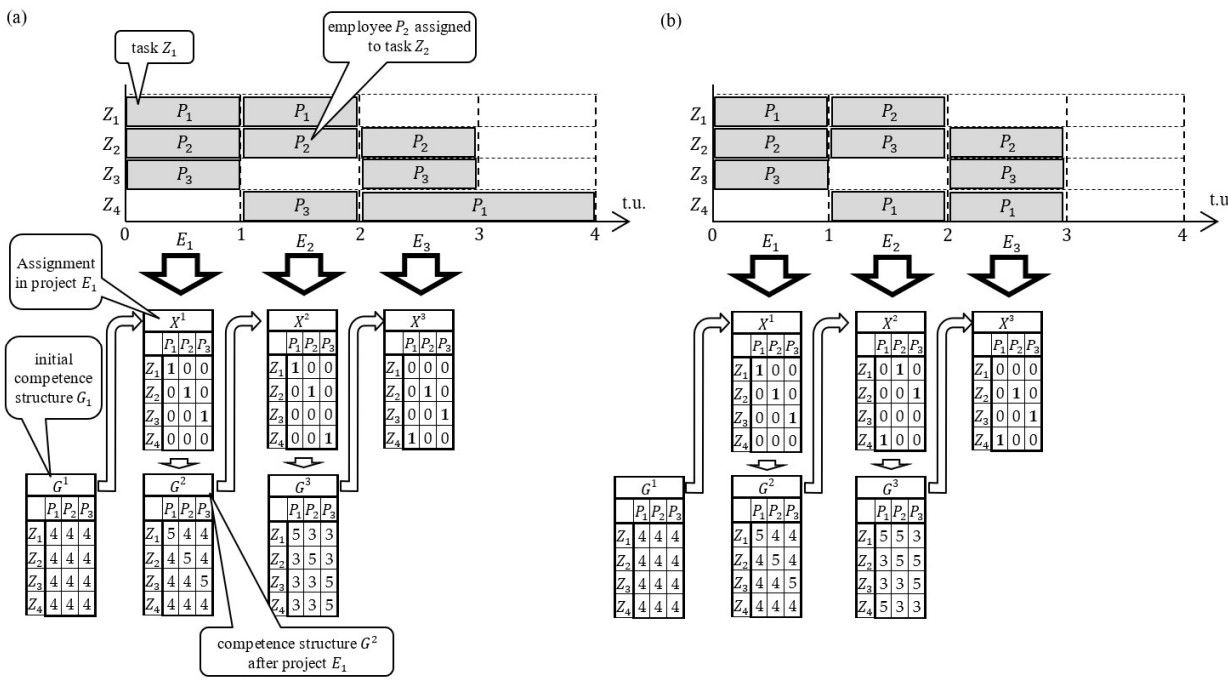

**Figure 1.** Project execution $E_1, E_2, E_3$ in horizon $H = 4$ (**a**), $H = 3$ (**b**).

However, the SPSP solutions found in the literature do not take into account unplanned events during the implementation of projects, e.g., additional orders. So, let us consider a case in which a company executes an order $E_3$ according to schedule $Y = (0, 1, 2)$ (see Figure 1b) and considers the possibility of accepting another order $E_2$. Therefore, the following question arises: is it possible to undertake the implementation of the $E_2$ project and complete the projects (in the order of $W = (E_1, E_2, E_3, E_2)$) within the deadline of 4 time units for the adopted allocation $X$ (determining schedule $Y$ from Figure 1b)? As can be seen from Figure 2, the $G^4$ competency structure allows for such an allocation of $X^4$ that results in a schedule $Y = (0, 1, 2, 3)$ that is completed on time.

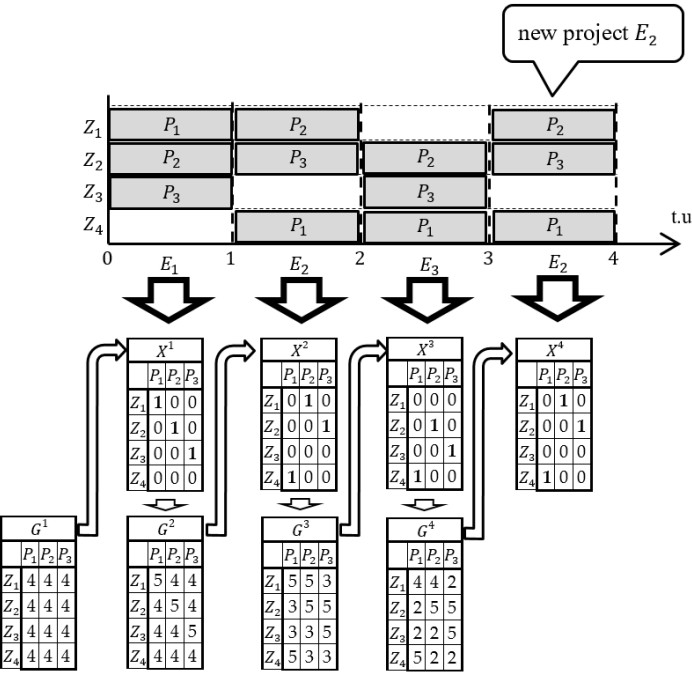

**Figure 2.** Variant of the schedule $Y = (0, 1, 2, 3)$ and execution of the projects $E_1, E_2, E_3, E_2$ in 4 time units.

## 4. Reference Model

The considered problem could be described using the declarative modeling paradigm.
Parameters:

$Z$: set of tasks $Z = \{Z_1, \ldots, Z_i, \ldots, Z_Q\}$;
$E$: portfolio of projects $E = \{E_1, \ldots, E_j, \ldots, E_K\}$;
$\mathcal{P}$: group of programmers $\mathcal{P} = \{P_1, \ldots, P_k, \ldots, P_M\}$;
$W$: sequence of projects;
$H$: expected moment (horizon) of completion of $W$ sequence projects;
$G^1$: initial competence structure;
$\varphi$: function determining the level of competence of programmers $\varphi\left(g_{i,k}^{j-1}, x_{i,k}^{j-1}\right)$;
$E^*$: set of disruptions $E^* \subseteq E$ (unplanned tasks/projects);
$^{(r)}W$: sequence of projects with disruption $^{(r)}E \in E^*$: $^{(r)}W = W||^{(r)}E$;
$R$: robustness of a competence structure;
$GR$: expected value of competence structure robustness for set of disruption $E^*$.

Decision variables:

$^{(r)}X^j$: assignment of programmers to tasks (2), in the case of disruption $^{(r)}E \in E^*$;
$^{(r)}G^j$: competence structure (1), in the case of disruption $^{(r)}E \in E^*$;
$^{(r)}Y$: schedule for project execution, in the case of disruption $^{(r)}E \in E^*$;
$^{(r)}\omega$: binary value specifying whether, in the case of disruption $^{(r)}E \in E^*$, the projects can be completed within the given horizon $H$.

Constraints:

- In each project $E_j$, for all tasks $Z_i$, the programmers of set $\mathcal{P}$ must be assigned the following:

$$\sum_{k=1}^{M} {}^{(r)}x_{i,k}^j = 1; \ \forall Z_i \in E_j, \ \forall^{(r)}E \in E^*, \tag{6}$$

$$\sum_{k=1}^{M} {}^{(r)}x_{i,k}^j = 0; \ \forall Z_i \notin E_j, \forall^{(r)}E \in E^*, \tag{7}$$

- The assignment of programmers to tasks of portfolio $E$ is the same for each case of disruptions $^{(r)}E, ^{(q)}E \in E^*$.

$$^{(r)}x_{i,k}^j = {}^{(q)}x_{i,k}^j; \ q \neq r, \ E_j \neq {}^{(r)}E, {}^{(q)}E \in E^* \tag{8}$$

- In each project $E_j$, programmer $P_k$ can only be assigned to one task.

$$\sum_{i=1}^{Q} {}^{(r)}x_{i,k}^j = 1; \ \forall Z_k \in \mathcal{P}, \ \forall^{(r)}E \in E^*, \tag{9}$$

- Elements $^{(r)}g_{k,i}^j$ of the competence structure $^{(r)}G^j$ of project $E_j$ (for $j > 1$) depend on the preceding project $E_{j-1}$.

$$^{(r)}g_{i,k}^j = \varphi\left({}^{(r)}g_{i,k}^{j-1}, {}^{(r)}x_{i,k}^{j-1}\right), \text{ for } j > 1, \ \forall^{(r)}E \in E^*, \tag{10}$$

- The duration of a task $^{(r)}t_i^j$ for the $E_j$ project depends on the competence of the $P_k$ employee who is performing it.

$$^{(r)}t_i^j = \begin{cases} 1 & \text{if } ^{(r)}x_{i,k}^j = 1 \text{ and } ^{(r)}g_{i,k}^j \geq 4 \\ 2 & \text{if } ^{(r)}x_{i,k}^j = 1 \text{ and } ^{(r)}g_{i,k}^j = 3, \ \forall Z_i \in E_j, \ \forall^{(r)}E \in E^*, \\ 4 & \text{if } ^{(r)}x_{i,k}^j = 1 \text{ and } ^{(r)}g_{i,k}^j \leq 2 \end{cases} \tag{11}$$

- Project $E_j$ starts after project $E_{j-1}$ is finished.

$$^{(r)}y_j = {}^{(r)}y_{j-1} + \max_{i=1...Q}\left\{{}^{(r)}t_i^{j-1}\right\}, \text{ for } j > 1, \; {}^{(r)}y_1 = 0, \; \forall^{(r)}E \in E^*, \tag{12}$$

- If the portfolio of projects (with additional project $^{(r)}E$) can be completed within the given horizon $H$, then the competence structure is robust for disruption $^{(r)}E$.

$$\left({}^{(r)}y_{K+1} + \max_{i=1...Q}\left\{{}^{(r)}t_i^{K+1}\right\} \leq H\right) \implies \left({}^{(r)}\omega = 1\right), \tag{13}$$

$$\left({}^{(r)}y_{K+1} + \max_{i=1...Q}\left\{{}^{(r)}t_i^{K+1}\right\} > H\right) \implies \left({}^{(r)}\omega = 0\right), \tag{14}$$

- Competence structure robustness $R$ (5) for the disruption of set $E^*$ should be at least equal to $GR$ as follows:

$$R = \frac{\sum_{r=1}^{|E^*|} {}^{(r)}\omega}{|E^*|} \geq GR. \tag{15}$$

The presented model allows us to answer the following question: is there an assignment $^{(r)}X$ that guarantees a robustness $R \geq GR$ for the given set of disruptions $E^*$? The above problem can be formulated as the following CSP (constraint satisfaction problem):

$$CS = ((\mathcal{V}, \mathcal{D}), \mathcal{C}), \tag{16}$$

where $\mathcal{V} = \left\{{}^{(r)}x_{i,k}^j, {}^{(r)}g_{i,k}^j, {}^{(r)}y_j \middle| k = 1, \ldots, M; i = 1, \ldots, Q; j = 1, \ldots, K+1, r = 1 \ldots |E^*|\right\}$, a set of decision variables representing assignment $^{(r)}X$, schedule $^{(r)}Y$ and competence structure $^{(r)}G^j$; $\mathcal{D}$ is a finite set of domains of the decision variables; and $\mathcal{C}$ is a set of constraints specified in inequalities (6)–(15).

To solve the problem *CS* (16), one should determine the values of decision variables $^{(r)}x_{i,k}^j, {}^{(r)}g_{i,k}^j, {}^{(r)}y_j$, for which all the constraints given in set $\mathcal{C}$ are satisfied. Solving *CS* means determining the assignment which guarantees the execution of the portfolio of projects with a given robustness for the occurrence of a new project $^{(r)}E \in E^*$.

The problem defined in this way should be treated as an extension of the worker assignment problem [59,60] (which is NP-hard) with elements of assessing the robustness of the solutions obtained. In this approach, in addition to the decision variables describing the allocation of employees $^{(r)}X^j$ (to which the classic assignment problem is limited), the level of their competencies $^{(r)}G^j$ as well as project implementation schedule $^{(r)}Y$ resulting from the adopted assignment are also taken into account. The search space determined by these variables grows exponentially with the number of programmers ($M$), implemented tasks ($Q$), and projects ($K$), as well as with the considered project implementation time horizon ($H$), and is estimated by the function $(M, Q, K, H) = 10^{M \times Q \times K} \times K^H$. The problem under consideration, similar to the worker assignment problem, is NP-hard. The use of declarative programming environments (in particular, constraint programming) allows for an effective search of such spaces using branch and bound [61] algorithms dedicated to solving the constraint satisfaction problem (CSP), implementing mechanisms of constraints propagation and variables distribution. For the *CS* problem (16) considered in the article, the IBM ILOG CPLEX declarative programming environment was used.

The proposed model describes the constraints coming from a software development company. The correctness of the model was verified by an IT industry expert in a series of 10 variants of test data from completed projects.

An example using the proposed approach is presented in the next section.

## 5. Computation Experiments

As part of the computer experiments, a case study related to the need to implement tasks in a software company was performed, and the effectiveness of the developed solution was checked for various scales of project/employee size.

### 5.1. Case Study

In order to verify the correctness of the model proposed in Section 4, an experiment was conducted based on data obtained from an enterprise specializing in the implementation of IT sector projects. For confidentiality reasons, it is impossible to provide the company's full name. In particular, the company develops IT programs and internet applications such as systems for launching multiple websites, intranet portals, browser and mobile applications, etc. Such a big variety of implemented projects requires specific competences and flexibility of human resources (programmers) to take over additional tasks when a specific order is received (multi-functional employees are required). Additionally, it is necessary to rotate the assignment of tasks to employees in order to maintain competences at a level that allows for effective project performance.

For the purposes of the experiments, data from the $E$ portfolio consisting of six projects were used as follows: $E = \{E_1, \ldots, E_6\}$, within which four tasks, $Z = \{Z_1, Z_2, Z_3, Z_4\}$, were carried out.

To carry out the tasks, the company employed staff comprising eight programmers $P = (P_1, \ldots, P_8)$. According to the adopted model, each employee was assessed in terms of the level of competence to perform individual tasks $Z = (Z_1, \ldots, Z_4)$. Their collective list is represented by the initial competence structure $G^1$ in Table 4. Due to the requirements for personal data protection, the processed data have been pseudonymized.

**Table 4.** Company's competence structure $G^1$.

| $G^1$ | $P_1$ | $P_2$ | $P_3$ | $P_4$ | $P_5$ | $P_6$ | $P_7$ | $P_8$ |
|-------|-------|-------|-------|-------|-------|-------|-------|-------|
| $Z_1$ | 5 | 4 | 5 | 4 | 3 | 4 | 4 | 5 |
| $Z_2$ | 4 | 4 | 5 | 4 | 4 | 5 | 3 | 4 |
| $Z_3$ | 4 | 4 | 5 | 4 | 5 | 5 | 4 | 4 |
| $Z_4$ | 5 | 5 | 4 | 4 | 5 | 4 | 5 | 3 |

The duration of tasks $Z_i$ depends on the competency level as follows:

- Duration is $t_i^j = 1$ time unit if the level of employee competency is 4 or 5;
- Duration is $t_i^j = 2$ time unit if the level of employee competency is 1, 2 or 3.

According to the learning effect, it is known that the competency level increases as follows:

- If the competency level is 1 or 2, the competence increases by one per two time units of the task duration;
- If the competency level is 3 or 4, the competence increases by one for each time unit of the task duration.

Similarly, according to the forgetting effect, it is known that competency level decreases as follows:

- If the competency level is 2 or 3, the competency decreases by one for each time unit when the task is not performed;
- If the competency level is 4 or 5, the competency decreases by one for two time units when the task is not performed.

Additionally assumed was the following:

- A single programmer, regardless of their competency level, can complete the task;
- Every task needs to be assigned to employees;

- The entire project portfolio $E = \{E_1, \ldots, E_6\}$ needs to be completed within the specified time horizon $H = 3$;
- A disruption in the form of an additional project $E^* = \{E_7\}$ which requires a set of tasks, $E_7 : (Z_1, Z_2, Z_3, Z_4)$, is adopted after the completion of the assumed portfolio $E$, which needs to be completed in time horizon $H = 4$.

The objective of the experiment is to utilize the constructed model in order to identify the assignment $^{(r)}X$ that guarantees robustness for given disruption $E^*$. Finding this kind of assignment provides a solution to the problem (16) formulated in the form of constraint programming and then is implemented in the declarative programming environment IBM ILOG CPLEX (Intel Core i7-M4800MQ 2.7 GHz, 32 GB RAM). The proposed sequence of assignments $X^1, X^2, X^3, X^4$ used to perform the $E$ project portfolio and additional project $E^*$ (disruption) is presented in Figure 3. This is one of the many variants of feasible solutions (which satisfies assumptions). The developed implementation of the model does not allow for the setting of optimal schedules, e.g., in terms of the smallest possible loss of competence. This issue will be the focus of future work.

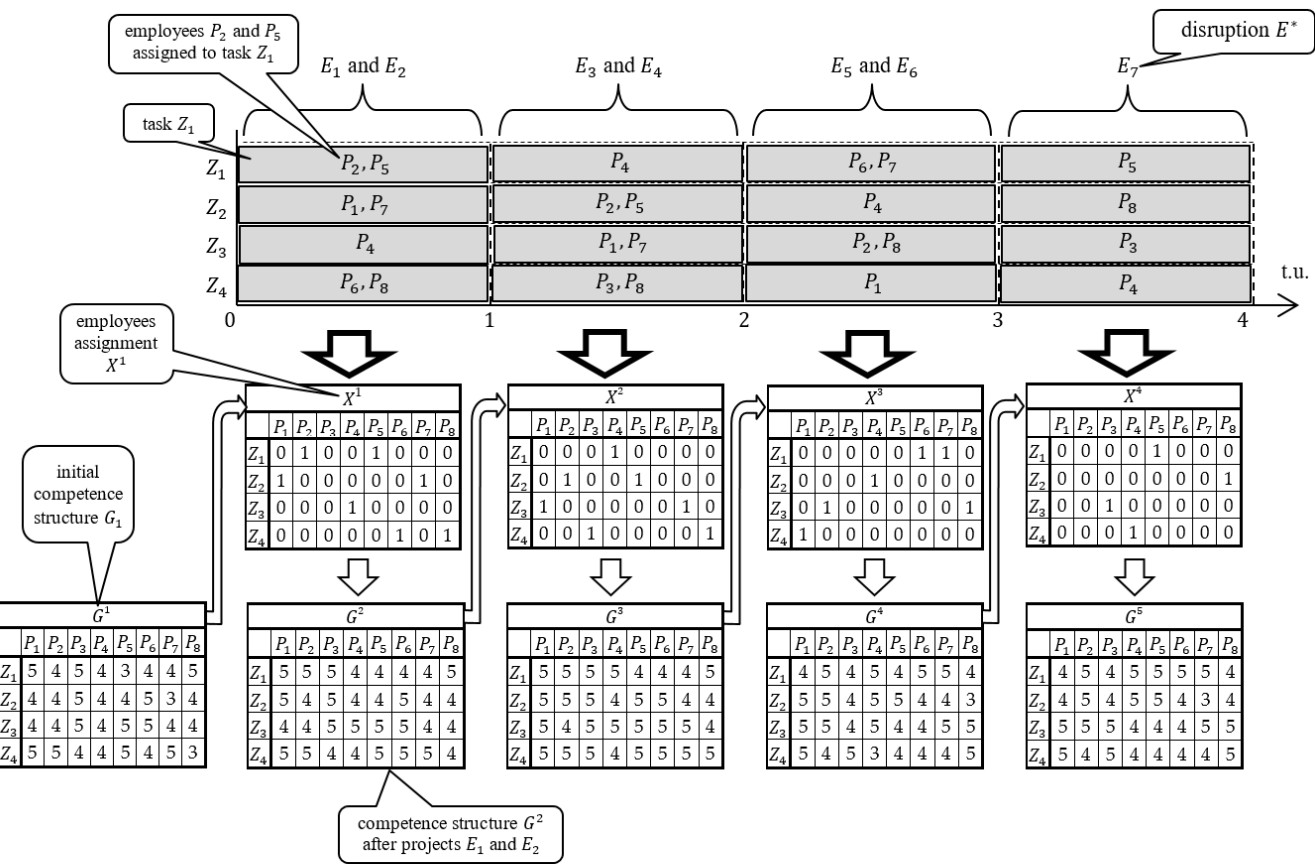

**Figure 3.** Sequence of an assignments $X^1, \ldots, X^4$ used to perform the $E$ project portfolio and disruption $E^*$.

### 5.2. Quantitative Calculations

The effectiveness of the proposed approach, which was determined by assessing its scalability, was validated through multiple computer experiments. The results, considering various numbers of programmers ($m = 10, 12, 15$) and tasks ($n = 20, \ldots, 50$), are illustrated in Figure 4.

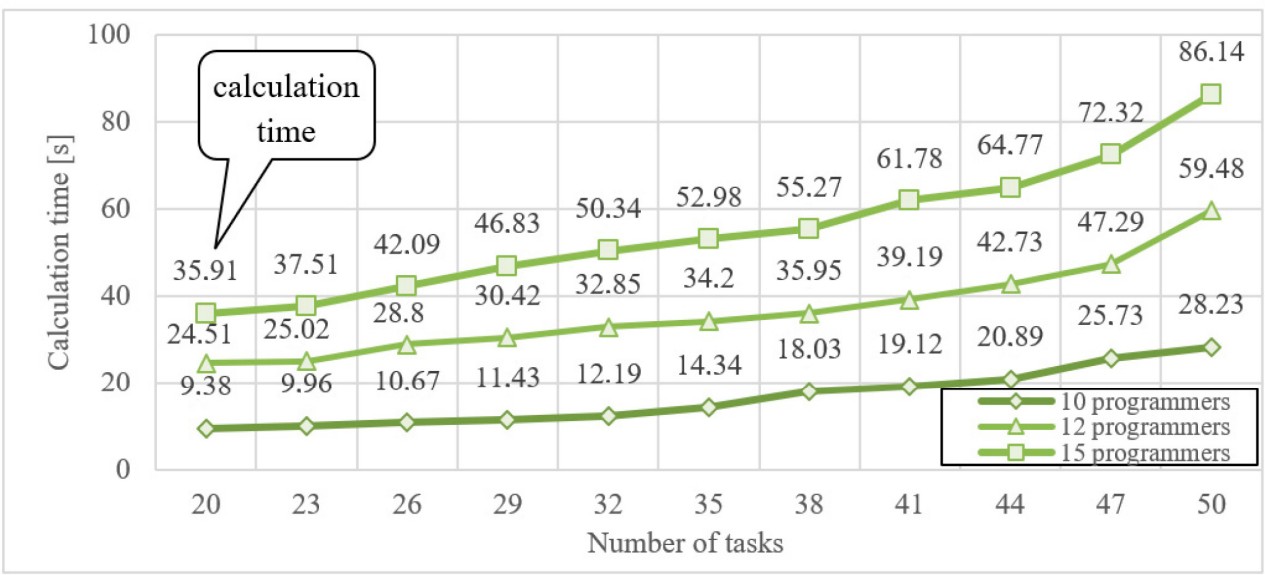

**Figure 4.** Calculation time for the various number of programmers.

In each experiment, project portfolio *E* consists of five projects, which include a variable total number of tasks (as mentioned above, from 20 to 50). The number of tasks in different (more or less equal) proportions was distributed among five projects (for example, for the variant of 20 tasks, they were divided into 4 tasks for each of the five projects, and for the variant of 23 tasks, they were divided into 4 tasks for the two projects and 5 tasks in the another three projects, etc.). The set of unplanned projects is a pool of five projects consisting of 10 tasks each, one of which is randomly selected and is considered to be disruption $E^*$. The sought solution to the problem should guarantee the robustness $R = 1$.

Notably, as the problem's scale increases, the computation time rises exponentially. As an illustration, consider the cases where there are 50 tasks (operations). It is evident that an increase in the number of programmers from 10 to 12 results in more than a 2-fold extension of the calculation time. Furthermore, enlarging the team size from 10 to 15 programmers leads to the calculation time growing more than 3-fold. Hence, the proposed approach is applicable online for practical scenarios within the limit of 50 tasks for the entire project portfolio and teams of up to 15 programmers.

*5.3. Experiments Summary*

In summarizing the case study and quantitative experiments, let us point out the following:

- When using the developed model, decision-makers can find a staffing rotation plan that ensures that competencies are maintained at a level that allows for effective project performance.
- The proposed model is suitable for the scale of problems that occur in real-life companies.
- The obtained computation times relate to finding the one admissible solution.
- The developed implementation of the model does not allow for the setting of optimal plans, e.g., in terms of the smallest possible loss of competencies and associated calculation times.
- The model does not contain sufficient conditions that guarantee the existence of non-empty sets of admissible solutions. This is one of the future research goals.

The used approach shows that the factors determining the need to introduce job rotation also include the number of activities performed, the planning horizon, and the initial level of competencies. This means that in some cases, e.g., due to a small number of projects, despite the use of rotation, it is not possible to avoid the effect of forgetting. However, the model we propose allows us to formulate synthesis questions, e.g.,

- What team of employees with what competencies makes it possible to use work rotation to avoid the forgetting effect?
- What set of orders (projects) will enable the team members to maintain the competencies at a given level?
- What allocation of team members with what competencies enables the use of work rotation to minimize losses caused by the forgetting effect?

## 6. Conclusions

The robust scheduling of a multi-skilled workforce through the job rotation approach is a strategic move that aligns with the demands of the modern business landscape. By fostering skill development, promoting adaptability, and enhancing employee satisfaction, organizations can create a workforce that is not only efficient but also well prepared for challenges of the future.

The paper introduces a new model aimed at proactively planning assignments of multi-skilled programmers to tasks within IT project portfolios while considering the forgetting effect. This approach addresses the challenge of maintaining the competency levels of programmers by advocating regular rotation between tasks.

A key advantage of this approach lies in its open declarative model structure, which facilitates the inclusion of new relationships between decision variables without compromising computational efficiency. In particular, in constraint programming environments such as IBM ILOG, an increase in constraints leads to a reduced solution determination time.

However, the limitations of this approach stem from specific workplace characteristics (such as task execution methods, group work dynamics, and resource usage) and the nature of disruptions encountered (e.g., employee absenteeism, competency loss, organizational structural changes). The model presented in this study is applicable solely to project portfolios defined by deterministic data. Using stochastic models that account for random or human-influenced processes shaping the project portfolio's implementation is unfeasible due to challenges in acquiring dependable random samples. Obtaining such samples is essential for identifying density distributions of random variables and synthesizing the stochastic parameters that govern the expected portfolio implementation. Moreover, practical disruptions encountered in real-world scenarios need consideration, such as the complete loss of qualifications (competencies), structural changes in task orders, and the simultaneous or consecutive absence of multiple employees, among others. Integrating these factors into the model remains an area for further exploration and development.

To overcome these limitations, future research will extend the model to consider uncertainties in operation times, planned deadlines for order execution, and skill levels expressed in fuzzy numbers. For example, exploring directed fuzzy numbers to describe imprecise variables in a constraints programming environment presents a promising approach. Additionally, our forthcoming work will focus on extending the model to accommodate the SPSP problem within dynamic environments. This expansion will consider unforeseen events such as the abrupt departure of employees and the subsequent recruitment of new team members with varying competencies. Expanding the broader context of our study, we aim to delve deeper into the following question: can a given allocation of multi-skilled team members guarantee a job rotation schedule that can maintain the current level of its competences? We plan to broaden this inquiry to encompass expectations related to minimizing team size while maximizing its robustness, a critical consideration for future studies.

The findings from this study carry significant managerial implications. Managers can use the proposed approach to dynamically rotate staff, ensuring the retention of their diverse skillsets without the need for costly additional training to refresh forgotten or outdated competencies. The case study demonstrates the applicability of this approach in industries where employees have a multitude of skills and where companies must adapt to evolving customer demands.

In industries that require employees to retain their skills for extended periods, rotation is emerging as a practical solution. The proposed approach helps managers in planning rotations for employee task assignments, enabling them to fulfill customer orders while maintaining an optimal competency structure within the team. This addresses the requirements of sustainable production, emphasizing effective human resource management by establishing clear protocols for planning, monitoring, and control.

It should be noted that continuous monitoring of existing competencies and supporting the development of new skills throughout an individual's career are necessary in the context of the challenges of the popular Industry 4.0 trend. It is particularly important to look for new methods to support decision-makers in maintaining the potential of human resources, and this is provided by our model.

**Author Contributions:** Conceptualization, E.S., G.B. and Z.B.; methodology, E.S. and G.B.; software, G.B.; validation, P.G.-D.; formal analysis, G.B.; investigation, E.S.; resources, P.G.-D. and E.S.; data curation, E.S. and P.G.-D.; writing—original draft preparation, Z.B. and E.S.; writing—review and editing, P.G.-D.; visualization, G.B. and E.S.; supervision, E.S.; project administration, E.S. and Z.B. All authors have read and agreed to the published version of the manuscript.

**Funding:** This research received no external funding.

**Data Availability Statement:** Data are contained within the article.

**Conflicts of Interest:** The authors declare no conflicts of interest.

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
