# Peer review of "Robust Scheduling of Multi-Skilled Workforce Allocation: Job Rotation Approach"

_electronics, doi:10.3390/electronics13020392_

Round 1
Reviewer 1 Report
Comments and Suggestions for Authors
The paper is very good and clear in my opinion and it addresses many important topics: the importance of job rotation, but also its effects on learning and forgetting. I only suggest few modifications to improve the paper before publication:
I suggest to add some references on Job Rotation, to let the reader understand that JR is not only beneficial to develop broad skills and to adapt changing demands, but it also may have benefits on the fatigue of the workers, balancing ergonomics indices. In the following, some papers you can consider to insert:
1)Caterino, M., Rinaldi, M., & Fera, M. (2023). Digital ergonomics: An evaluation framework for the ergonomic risk assessment of heterogeneous workers. International Journal of Computer Integrated Manufacturing, 36(2), 239-259.
2)Moussavi, S. E., Zare, M., Mahdjoub, M., & Grunder, O. (2019). Balancing high operator's workload through a new job rotation approach: Application to an automotive assembly line. International Journal of Industrial Ergonomics, 71, 136-144.
3)Otto, A., & Battaïa, O. (2017). Reducing physical ergonomic risks at assembly lines by line balancing and job rotation: A survey. Computers & Industrial Engineering, 111, 467-480.
I suggest to add a disucssion section before conclusions to discuss the results of your paper and the differences/similarities with the existing literature you have reviewed in the paper
Comments on the Quality of English Language
The English is ok. I noticed just few grammar errors. Please read the paper again and correct them.
Author Response
The authors thank the reviewers for their very helpful comments and suggestions. The authors have incorporated the reviewers’ comments in the revised manuscript.
The paper is very good and clear in my opinion and it addresses many important topics: the importance of job rotation, but also its effects on learning and forgetting. I only suggest few modifications to improve the paper before publication:
Remark 1: I suggest to add some references on Job Rotation, to let the reader understand that JR is not only beneficial to develop broad skills and to adapt changing demands, but it also may have benefits on the fatigue of the workers, balancing ergonomics indices. In the following, some papers you can consider to insert:
- Caterino, M., Rinaldi, M., & Fera, M. (2023). Digital ergonomics: An evaluation framework for the ergonomic risk assessment of heterogeneous workers. International Journal of Computer Integrated Manufacturing, 36(2), 239-259.
- Moussavi, S. E., Zare, M., Mahdjoub, M., & Grunder, O. (2019). Balancing high operator's workload through a new job rotation approach: Application to an automotive assembly line. International Journal of Industrial Ergonomics, 71, 136-144.
- Otto, A., & Battaïa, O. (2017). Reducing physical ergonomic risks at assembly lines by line balancing and job rotation: A survey. Computers & Industrial Engineering, 111, 467-480.
Response: Thank you very much for your valuable remark concerning papers on the benefits of job rotation. In Section 2.1. we have already mentioned the benefits (lines 91-92: ‘Job rotation, also known as work rotation, represents a managerial tactic employed within organizations to alleviate work monotony, boredom, fatigue, and burnout’). However, thanks to the reviewer's attention, we have added two from the above mentioned most recent articles (references [28] and [31]). Moreover, we also placed a concept of balancing ergonomics indices and a reference to the indicated literature (lines 93-94: ‘Its primary objective extends to improving job satisfaction, employee motivation and balancing ergonomics indices.’)
Remark 2: I suggest to add a discussion section before conclusions to discuss the results of your paper and the differences/similarities with the existing literature you have reviewed in the paper.
Response: Thank you for this valuable suggestion. Taking it into account, an additional discussion was held, covering similarities to existing works and emphasizing new aspects related to the possibility of formulating and solving synthesis problems, i.e. seeking answers to questions such as: what values ​​characterizing a given problem guarantee the existence of its acceptable solution? The paragraph we added is included at the end of Section 5:
Page 15, lines 571-581: ‘The used approach shows that the factors determining the need to introduce job rotation also include the number of activities performed, the planning horizon and the initial level of competencies. This means that in some cases, e.g. due to a small number of projects, despite the use of rotation, it is not possible to avoid the effect of forgetting. However, the model we propose allows us to formulate synthesis questions:
- What team of employees with what competencies makes it possible to use work rotation to avoid the forgetting effect?
- What set of orders (projects) will enable the team members to maintain the competencies at a given level?
- What allocation of team members with what competencies allows the use of work rotation to minimize losses caused by the forgetting effect?’
Remark 3: The English is ok. I noticed just few grammar errors. Please read the paper again and correct them.
Response: Thank you for drawing our attention to the linguistic aspect. The work has been proofread again.

Reviewer 2 Report
Comments and Suggestions for Authors
The article introduces an innovative model for task planning in programming teams, addressing the challenge of scheduling IT projects and proposing task rotation as a solution to optimize resources and enhance productivity in dynamic business environments.
The introduction offers a thorough background, highlighting organizational challenges and emphasizing the potential benefits of task rotation. The reference model for the Skill-based Project Scheduling Problem (SPSP) is presented as a Constraint Satisfaction Problem (CSP), providing a formal and structured approach.
The strengths of the article lie in the formalization using a declarative modeling paradigm, a clear definition of parameters and decision variables, and well-defined constraints ensuring feasibility. The model aims to find assignments guaranteeing competence structure robustness in disruptions.
Areas for improvement include addressing the computational complexity of the proposed CSP, discussing practical implementation considerations, providing details on solution approaches, and outlining validation methods. The article lacks explicit discussion on the computational complexity of solving the CSP, a crucial consideration for larger problem instances. Furthermore, practical constraints and real-world factors are mentioned as excluded, and it is essential to explore how these could be incorporated.
The article does not provide information on how the proposed model will be validated or tested, which is crucial for ensuring accurate representation and meaningful results.
The case study is presented clearly, demonstrating the effectiveness of the proposed approach in various scenarios. The conclusions align with the objectives, emphasizing the strategic importance of robust scheduling in a multi-skilled workforce.
In summary, the article provides valuable insights into proactive planning of assignments for multi-skilled programmers, presenting a well-structured model with clear results. However, addressing practical constraints and providing further details on solution approaches would significantly enhance the overall contribution of the research.
Author Response
The authors thank the reviewers for their very helpful comments and suggestions. The authors have incorporated the reviewers’ comments in the revised manuscript.
Remark 1: Areas for improvement include addressing the computational complexity of the proposed CSP, discussing practical implementation considerations, providing details on solution approaches, and outlining validation methods. The article lacks explicit discussion on the computational complexity of solving the CSP, a crucial consideration for larger problem instances. Furthermore, practical constraints and real-world factors are mentioned as excluded, and it is essential to explore how these could be incorporated.
Response: Thank you for this comment. The issue of allocating employees (in our case programmers) to tasks pending execution, project stages, work positions, etc. is most often formulated in literature as employee timetabling or staff scheduling [59, 60]. Such problems refer to the finding of work schedules for an organization’s staff that allow for the performance of a specific set of orders, services, projects, etc. There are many varieties of this problem, such as university timetabling problem, nurse scheduling problem, crew scheduling problem, etc. The examined problems may differ by objective functions, constraints, etc. In practice, on account of the discrete and combinatorial character of the above-listed issues, these are NP-hard problems.
The problem considered in this work should be treated as an extension of the worker assignment problem with elements of assessing the robustness of the obtained solutions. In this approach, in addition to the decision variables describing the allocation of employees (rX^j) (to which the classic assignment problem is limited), the level of their (rG^j) competencies, as well as the schedule for the implementation of the (rY) project resulting from the adopted assignment, are also taken into account. The search space determined by these variables grows exponentially with the number of: programmers (M), implemented tasks (Q), projects (K) as well as with the considered project implementation time horizon (H) and is estimated by the function: f(M,Q,K,H)=〖10〗^(M×Q×K)×K^H. However, the use of declarative programming environments (in particular constraint programming) allows for an effective search of such spaces using branch and bound [61] algorithms dedicated to solving constraint satisfaction problem (CSP), implementing mechanisms of constraints propagation and variables distribution. For the needs of the considered CS problem (16), the IBM ILOG CPLEX declarative programming environment was used. To summarize our response, in order to expand the manuscript to include discussion related to computational complexity, the article has been supplemented with the following paragraph:
Page 11-12, lines 448-463: The problem defined in this way should be treated as an extension of the worker assignment problem [59,60] (which is NP-hard) with elements of assessing the robustness of the solutions obtained. In this approach, in addition to the decision variables describing the allocation of employees (rX^j) (to which the classic assignment problem is limited), the level of their competencies (rG^j) as well as project implementation schedule (rY) resulting from the adopted assignment are also taken into account. The search space determined by these variables grows exponentially with the number of: programmers (M), implemented tasks (Q), and projects (K) as well as with the considered project implementation time horizon (H), and is estimated by the function: f(M,Q,K,H)=〖10〗^(M×Q×K)×K^H. The problem under consideration, similarly to the worker assignment problem, is NP-hard. The use of declarative programming environments (in particular, constraint programming) allows for an effective search of such spaces using branch and bound [61] algorithms dedicated to solving constraint satisfaction problem (CSP), implementing mechanisms of constraints propagation and variables distribution. For the CS problem (16) considered in the article, the IBM ILOG CPLEX declarative programming environment was used.
[59] Ernst, A.T.; Jiang, H.; Krishnamoorthy, M.; Sier, D. Staff Scheduling and Rostering: A Review of Applications, Methods and Models. Eur J Oper Res 2004, 153, 3–27, doi:10.1016/S0377-2217(03)00095-X.
[60] Panik, M.J. Linear Programming and Resource Allocation Modeling; Wiley, 2018; ISBN 9781119509448.
[61] Pesant, G. From Support Propagation to Belief Propagation in Constraint Programming. Journal of Artificial Intelligence Research 2019, 66, doi:10.1613/jair.1.11487
Remark 2: The article does not provide information on how the proposed model will be validated or tested, which is crucial for ensuring accurate representation and meaningful results.
Response: Thank you for your comment. The proposed model describes the constraints coming from a software development company. The correctness of the model was verified by an IT industry expert in a series of 10 variants of test data from completed projects. Therefore, we have added the following paragraph to the manuscript:
Page 12, lines 464-466: ‘The proposed model describes the constraints coming from a software development company. The correctness of the model was verified by an IT industry expert in a series of 10 variants of test data from completed projects.’
Remark 3: In summary, the article provides valuable insights into proactive planning of assignments for multi-skilled programmers, presenting a well-structured model with clear results. However, addressing practical constraints and providing further details on solution approaches would significantly enhance the overall contribution of the research.
Response: Thank you for this suggestion related to Remark 1, in response to which we have conducted a relevant discussion providing details on practical implementation.

Reviewer 3 Report
Comments and Suggestions for Authors
The article proposes an approach for multi-skilled workforce allocation including job rotation to address the software project scheduling problem.
The introduction is well-written and provides a comprehensive overview of the problem and existing literature.
The problem formulation is clear and good to understand, with well-illustrated examples. However, there are some minor issues in the problem formulation that should be addressed and improved:
· In the definition of gk,I (line 228) it is not clear what is the meaning of the values 1, …, 5. Later, when the example is introduced, it gets clear. However, it is advisable to provide an explanation of the meaning the first time it is mentioned.
· What is the meaning of the EY variable (line 238) and the H variable (line 263)? Please provide clear definitions.
· The notation of the robustness is a little bit confusing. In the text it is denoted as CSR (line 268), in the formula as R (line 271), GR in the parameter list (line 397) and w in the list of decision variables (line 403). Please use the same variable for the same meaning and explain the difference if another symbol is introduced.
· Table 1: The notation of the matrix G1 is not in line with the definition of Gj (formula 1). The first index k (programmer) should refer to the row and the second index i (task) to the column. But the table is shown in a transposed way.
· Table 3: The name of the matrix should be G2, not G0.
· Please explain the meaning of Equation 8 of the constraints (line 409). The description in line 406 is too short for explaining the following 3 equations.
· In line 422 the value of w defined if the formulated inequality is met. What is the value of w if the inequality is not met?
The description of the case study is clear, but the explanation of the Quantitative calculations in section 5.2 is too short. It is mentioned that there are 5 projects, each with 10 tasks. How are the calculations done with number of tasks less than 50 (see Figure 4). What are the 5 scenarios mentioned in line 514 and why is R = 1?
Author Response
The authors thank the reviewers for their very helpful comments and suggestions. The authors have incorporated the reviewers’ comments in the revised manuscript.
Remark 1: In the definition of gk,I (line 228) it is not clear what is the meaning of the values 1, …, 5. Later, when the example is introduced, it gets clear. However, it is advisable to provide an explanation of the meaning the first time it is mentioned. Response: Thank you for this recommendation. The values 1,…,5 of the g_(k,i)^j variable determines the level of competence of the programmer P_k necessary to perform task Z_i. The article adopts a five-level scale, the values of which determine the duration time t_i^j of task Z_i. For clarification purposes, the necessary description has been added in the following paragraph: Page 5, lines 227-234: “where g_(i,k)∈{1,…,5} determines the level of competence of the programmer P_k necessary to perform task Z_i. The adopted five-level scale determines the duration time t_i^j of task Z_i in accordance with the following function: g_(i,k)^j={((5 or 4&"programmer " P_k " performs the task " Z_i in t_i^j=1 time unit 3&"programmer " P_k " performs the task " Z_i in t_i^j= 2 time units) "2 or 1 programmer " P_k " performs the task " Z_i in t_i^j= 4 time units) According to the above, the duration time of task Z_i is: 1 time unit if the programmer P_k has competences at level 4 or 5, 2 time units if the programmer P_k has competences at level 3, 4 time units if the programmer P_k has competences at level 2 or 1.” Remark 2: What is the meaning of the EY variable (line 238) and the H variable (line 263)? Please provide clear definitions. Response: Thank you for your remark. The variable EY denotes the deadline for completing the Y schedule of projects defined by the sequence W. In turn, H denotes the given horizon for completing the sequence of projects W. In other words, the notation EY≤H means that the deadline EY to complete the sequence of projects W determined by the schedule Y must be less than or equal to the arbitrarily set horizon H for performing the sequence of projects W. To more precisely define the symbols used, the parameter H was added to the model (page 10, line 399: H: expected moment (horizon) of completion of W sequence projects) and corrected the relevant rows: Page 6, lines 244-245: ‘The completion time of project sequence order W depends on schedule Y and is denoted as variable EY.’ Page 6, lines 270-271: ‘The moment EY of completing sequence projects W determined by the schedule Y must be less than or equal to the arbitrarily set horizon H (EY≤H).’ Remark 3: The notation of the robustness is a little bit confusing. In the text it is denoted as CSR (line 268), in the formula as R (line 271), GR in the parameter list (line 397) and w in the list of decision variables (line 403). Please use the same variable for the same meaning and explain the difference if another symbol is introduced. Response: Of course, you are right. We have removed CSR and used only two symbols, GR and R, where R is the robustness value of the obtained solution and GR is the expected value of robustness, i.e. R≥GR: Page 6, line 274-275: ‘The ability of the competency structure to deal with this type of disruption can be assessed by the robustness measure.’ Page 10, line 410-411: ‘rω: binary value specifying whether in the case of disruption rE∈E* the projects can be completed within the given horizon H.’ Page 10, line 404: ‘R: robustness of a competence structure.’ Remark 4: Table 1: The notation of the matrix G1 is not in line with the definition of Gj (formula 1). The first index k (programmer) should refer to the row and the second index i (task) to the column. But the table is shown in a transposed way. Response: Thank you for your valid remark. To match the notation in the table, we have changed the order of indexes k and i. Remark 5: Table 3: The name of the matrix should be G2, not G0. Response: Thank you. Mistake has been corrected. Remark 6: Please explain the meaning of Equation 8 of the constraints (line 409). The description in line 406 is too short for explaining the following 3 equations. Response: Thank you for your comment. The constraint (8) means that the assignment of programmers to tasks of portfolio E is the same for each case of disruptions rE,qE∈E*. Therefore, we have corrected it and added the following explanation: Page 11, lines 416-418: The assignment of programmers to tasks of portfolio E is the same for each case of disruptions {Expression} (8) - Please see the attachment. Remark 7: In line 422 the value of w defined if the formulated inequality is met. What is the value of w if the inequality is not met? Response: Thank you for your valid remark. In the case when the inequality is not met then rω=0. Thus, we have added the following constraint: Page 11, line 432: {Expression} (14) - Please see the attachment. Remark 8: The description of the case study is clear, but the explanation of the Quantitative calculations in section 5.2 is too short. It is mentioned that there are 5 projects, each with 10 tasks. How are the calculations done with number of tasks less than 50 (see Figure 4). What are the 5 scenarios mentioned in line 514 and why is R = 1? Response: Thank you for your valid remark. The description of the experiment is not very precise and we also incorrectly indicated that there are 10 tasks in each of the 5 projects. Of course, we tested different variants for a variable total number of tasks (from 20 to 50), which were distributed in different (more or less equal) proportions to 5 projects (for example, for the variant of 20 tasks, they were divided into 4 tasks for each of the 5 projects). We also did not discuss 5 scenarios of unplanned projects and how to mark them as R=1. We wanted to look for solutions to the problem for which the robustness would be at the level of R=1. However, 5 scenarios of unplanned projects create a pool of projects consisting of 10 tasks each, of which a random one is considered as a disruption. To better explain this issue, we have introduced the following paragraph in Section 5.2: Page 14-15, lines 543-550: ‘In each experiment, project portfolio E consists of 5 projects, which include a variable total number of tasks (as mentioned above: from 20 to 50). The number of tasks in different (more or less equal) proportions was distributed among 5 projects (for example: for the variant of 20 tasks, they were divided into 4 tasks for each of the 5 projects, for the variant of 23 tasks, they were divided into 4 tasks for the 2 projects and 5 tasks in another 3 projects, etc.). The set of unplanned projects is a pool of 5 projects consisting of 10 tasks each, one of which is randomly selected and is considered as a disruption E*. The sought solution to the problem should guarantee the robustness R=1.’

Reviewer 4 Report
Comments and Suggestions for Authors
Thank you very much for the opportunity to review this interesting paper. Below, I am going to suggest some important issues to improve this paper.
Briefly summarize the key findings of the experiment and how they support the model's effectiveness. Provide concrete examples of workplace characteristics and disruptions that may limit the model's applicability. Clearly define the research questions and expected outcomes of the proposed future work with fuzzy numbers and uncertainties. Conclude by emphasizing the model's potential impact and contributions to the field of project scheduling and resource management. Require proofreading with the words "so-called, etc." mentioned in the paper. If a software SPSP has a name, it can't be the so-called name. Other issues with the proofreading are obvious. Moreover, There are certain issues in the conclusion: Robust scheduling lacks specific details about the proposed model. Is there a link between the model and the findings? The list of limitations mentions workplace characteristics and disruptions but lacks concrete examples or specific issues identified in the research. This makes it difficult to understand the scope and practical implications of these limitations. While mentioning fuzzy numbers for skill levels and operational uncertainties, the description of future research lacks a clear direction or specific research questions. I hope authors will address these issues and solve them perfectly.Author Response
The authors thank the reviewers for their very helpful comments and suggestions. The authors have incorporated the reviewers’ comments in the revised manuscript.
Remark 1: Briefly summarize the key findings of the experiment and how they support the model's effectiveness.
Response: Thank you for your suggestion. To summarize the experiments, we added Section 5.3. "Experiments summary", which contains the following paragraph:
Page 15, lines 558-570: ‘Summarizing the case study and quantitative experiments, let us point out:
- When using the developed model, decision-makers can find a staffing rotation plan that ensures maintaining competencies at a level that allows for effective project performance.
- The proposed model is suitable for the scale of problems that occur in real-life companies.
- The obtained computation times relate to finding the one admissible solution.
- The developed implementation of the model does not allow setting optimal plans, e.g. in terms of the smallest possible loss of competencies and associated calculation times.
- The model does not contain sufficient conditions that guarantee the existence of non-empty sets of admissible solutions. This is one of the future research goals.’
Remark 2: Provide concrete examples of workplace characteristics and disruptions that may limit the model's applicability.
Response: Thank you for this suggestion. To provide examples of workplace characteristics and disruptions that may limit the model's applicability (located in Conclusions in lines 597-500: „However, the limitations of this approach stem from specific workplace characteristics (such as task execution methods, group work dynamics, and resource usage) and the nature of disruptions encountered (e.g., employee absenteeism, competency loss, organizational structural changes”) we extended by the following paragraph:
Page 16, lines 600-609: ‘The model presented in this study is applicable solely to project portfolios defined by deterministic data. Using stochastic models that account for random or human-influenced processes shaping the project portfolio's implementation is unfeasible due to challenges in acquiring dependable random samples. Obtaining such samples is essential for identifying density distributions of random variables and synthesizing the stochastic parameters that govern the expected portfolio implementation. Moreover, practical disruptions encountered in real-world scenarios need consideration, such as: complete loss of qualifications (competencies), structural changes in task orders, and simultaneous or consecutive absence of multiple employees, among others. Integrating these factors into the model remains an area for further exploration and development.’
Remark 3: Clearly define the research questions and expected outcomes of the proposed future work with fuzzy numbers and uncertainties.
Response: Thank you very much for this comment. The topic of future research on uncertainties and fuzzy numbers (located in Conclusions in lines 610-612: ‘To overcome these limitations, future research will extend the model to consider uncertainties in operation times, planned deadlines for order execution, and skill levels expressed in fuzzy numbers.’) have been expanded by the following paragraph:
Page 16, lines 612-622: ‘For example, exploring directed fuzzy numbers to describe imprecise variables in a constraints programming environment presents a promising approach. Additionally, our forthcoming work will focus on extending the model to accommodate the SPSP problem within dynamic environments. This expansion will consider unforeseen events such as abrupt departure of employees and the subsequent recruitment of new team members with varying competencies. Expanding the broader context of our study, we aim to delve deeper into the question: Can a given allocation of multi-skilled team members guarantee a job rotation schedule that can maintain the current level of its competences? We plan to broaden this inquiry to encompass expectations related to minimizing team size while maximizing its robustness, a critical consideration in future studies.’
Remark 4: Conclude by emphasizing the model's potential impact and contributions to the field of project scheduling and resource management.
Response: Thank you for this suggestion. It was well noticed that the conclusions did not indicate the impact of the developed model on practical use. Therefore, an appropriate paragraph has been added:
Page 16, lines 623-639: ‘The findings from this study carry significant managerial implications. Managers can use the proposed approach to dynamically rotate staff, ensuring the retention of their diverse skill sets without need for costly additional training to refresh forgotten or outdated competencies. The case study demonstrates the applicability of this approach in industries where employees have a multitude of skills and companies must adapt to evolving customer demands.
In industries that require employees to retain their skills for extended periods, rotation is emerging as a practical solution. The proposed approach helps managers in plan rotations for employee task assignments, enabling them to fulfill customer orders while maintaining an optimal competency structure within the team. This addresses the requirements of sustainable production, emphasizing effective human resource management by establishing clear protocols for planning, monitoring, and control.
It should be noted that continuous monitoring of existing competencies and supporting the development of new skills throughout an individual's career are necessary in the context of the challenges of the popular Industry 4.0 trend. It is particularly important to look for new methods to support decision makers in maintaining human resources potential, and this is provided by our model.’
Remark 5: Require proofreading with the words "so-called, etc." mentioned in the paper. If a software SPSP has a name, it can't be the so-called name. Other issues with the proofreading are obvious.
Response: Thank you for drawing attention to this aspect. The indicated issues have been corrected and the work has been proofread again.

Round 2
Reviewer 2 Report
Comments and Suggestions for Authors
I appreciate the authors' thorough response to the comments and suggestions raised during the review process. The incorporation of additional information on computational complexity, practical implementation considerations, and validation methods has significantly enhanced the manuscript.
I am now satisfied with the revisions made by the authors, addressing the concerns outlined in Remark 1 and Remark 2. The expanded discussion on the computational complexity of the proposed Constraint Satisfaction Problem (CSP), coupled with the inclusion of details regarding the IBM ILOG CPLEX declarative programming environment, contributes to a more comprehensive understanding of the research.
Moreover, the clarification on model validation through testing with 10 variants of real-world data from completed projects provides a solid foundation for the reliability of the proposed model, addressing the concerns raised in Remark 2.
The detailed discussion provided in response to Remark 1, along with the added paragraph on model validation in response to Remark 2, aligns well with the recommendations for addressing practical constraints and providing further details on solution approaches, as mentioned in Remark 3.
In light of these revisions, I recommend accepting the manuscript for publication.
Reviewer 4 Report
Comments and Suggestions for Authors
The authors address my suggestions and comments very well in the revised version of the manuscript. I am happy with their work and would like to congratulate them for their hard work on this paper.